# EVALUATING LLM GOAL-DIRECTEDNESS

## ABSTRACT

To what extent do LLMs use their capabilities towards their given goal? We take this as a measure of their *goal-directedness*. We evaluate goal-directedness on tasks that require information gathering, cognitive effort, and plan execution, where we use subtasks to infer each model's relevant capabilities. Our evaluations of LLMs from Google DeepMind, OpenAI, and Anthropic show that goal-directedness is relatively consistent across tasks, differs from task performance, and is only moderately sensitive to motivational prompts. Notably, most models are not fully goal-directed. We hope our goal-directedness evaluations will enable better monitoring of LLM progress, and more deliberate design choices of agentic properties in LLMs.

## 1 INTRODUCTION

Large Language Models (LLMs) are increasingly used to perform complex tasks that require multiple capabilities to be combined towards a larger goal. Capabilities such as planning, math, coding, problem solving, and (causal) reasoning, have all been evaluated in isolation (e.g. Hao et al., 2024; Huang et al., 2024; Ahn et al., 2024). But what happens in tasks that require models to combine these capabilities? We find that performance often deteriorates. For example, most models are reasonably good at estimating the height of a block from noisy measurements when this is their only task, but much worse at it when it's part of a larger task (see Figure 1 which shows that as part of a larger task, LLMs fail to fully employ this capability.).

We define a LLM model's **goal-directedness** as its *propensity to use available resources and capabilities to achieve a given goal*. Goal-directedness is a key agentic property (Dennett, 1989; Dung, 2024), and important to understand for multiple reasons. First, more goal-directed LLMs can likely form more autonomous agents. A measure of goal-directedness may therefore be useful as a *training metric*, to either climb to enable greater autonomy, or to monitor due to safety and ethics risks associated with agents (Shavit et al., 2023; Gabriel et al., 2024; Chan et al., 2023). In particular, goal-directedness is a prerequisite for AIs employing unethical or unsafe means in pursuit of a longer-term goal. Such *convergent instrumental subgoals* (Omohundro, 2018) are a key ingredient in many threat models from AI systems (Kenton et al., 2022), ranging from resource acquisition (Bostrom, 2014), deception (Greenblatt et al., 2024), preference manipulation (Carroll et al.,

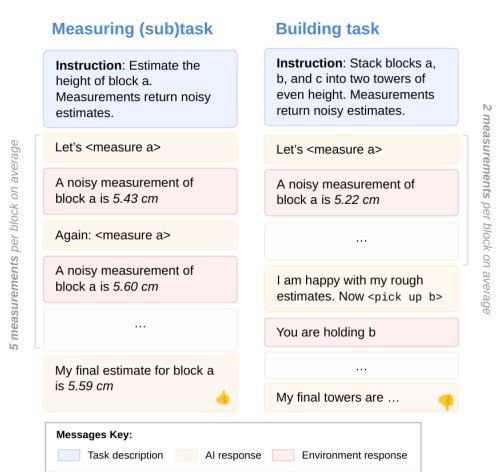

Figure 1: How motivated are LLMs to do their tasks well? Do they sometimes slack off?

2021), inappropriate relationships (Gabriel et al., 2024), sycophancy (Sharma et al., 2023), or power-seeking (Carlsmith, 2022). Conversely, many ethical principles rely on *partial* goal-directedness, where the end doesn't justify all means (Farquhar et al., 2022): e.g. it's good to make money, but not by fraud. Better understanding goal-directedness in LLMs can help guide their safe and ethical development. Finally, goal-directedness is an important component of human psychology (American Psychological Association, 2024), and it is of scientific interest to see how it carries over to LLMs.

Goal-directedness has been studied in animals, humans, and AI (Section 2). However, none of the past approaches are directly applicable to LLMs, as they are typically real-time, and frequently audio-visual (Tinius, 2003). Meanwhile, approaches in AI often equate goal-directedness with task performance (Shimi et al., 2021; MacDermott et al., 2024), ignoring differences in capability. But task performance arguably says more about capability than goal-directedness. Rather, it is the (un)willingness to employ a capability that measures goal-directedness, as illustrated in Figure 1.

**Key contributions** Our first key contribution is a formal definition of (capability-conditioned) goal-directedness GD applicable to LLMs (Section 3.1). Computing GD requires knowing a model's capabilities, which we infer from subtask performance. GD differs empirically from task performance (Section 4.2), and is consistent across tasks, and with other measures of GD (Section 4.3). Our second key contribution is a framework for evaluating GD across four main tasks.[1] The tasks involve information gathering, cognitive effort, and plan execution. We evaluate state-of-the-art models from Google DeepMind, Anthropic, and OpenAI. Third, most models are not fully goal-directed, i.e. they fail to fully apply their capabilities in some tasks, especially information gathering (Section 4.1). Motivational prompting only helps somewhat (Section 4.4). We discuss limitations in Section 5.

## 2 RELATED WORK

**Human goal-directedness** According to the American Psychological Association (American Psychological Association, 2024), (human) goal-directed behavior is oriented towards attaining a particular goal, and consists of purposeful and deliberate actions. Unlike habitual or reflexive behavior which happens automatically or instinctively and is relatively insensitive to the value of behavioral goals, goal directed behavior selects actions according to their outcomes (Pezzulo et al., 2014; Steinglass & Foerde, 2016). Hallmarks of goal-directedness are the capacity to evaluate consequences of actions, maintain behavior consistent with the goal, focus on relevant information, and ignore distractions (Miller & Wallis, 2009; Bunge & Souza, 2009; Phelps & Russell, 2023). In general, humans are more likely to commit to a goal when they positively evaluate its value (Locke & Latham, 2019). Goal-directedness is related to motivation: a motivated person is more likely to set goals and engage in pursuing them.

Tests for measuring human GD and motivation include *progress ratio tasks* (Chen et al., 2022; Wolf et al., 2014), where subjects must complete increasingly large task to get another (fixed-size) reward, and the *anagram persistence test* (Gignac & Wong, 2020), where subjects need to create real words with a given set of letters (sometimes no word can be created at all). For both tests, how long subjects persist in trying to solve the problem is indicative of goal-directedness. Other tasks include measures of sustained and selective attention such as *continuous performance tasks* (Tinius, 2003), measures of inhibitory control such as *go/no-go tasks* (Gomez et al., 2007), the *stop signal task* (PsyToolkit, 2024) and the *Stroop test* (Wikipedia, 2024), assessments of the cognition behind the action such as *instrumental devaluation* (Mannella et al., 2016), as well as *questionnaires* on self-reported motivation (Center for Self-Determination Theory, 2024).

**AI goal-directedness** Goal-directedness has been explored from a few different angles in the field of AI. A *goal-oriented task* is characterized as requiring sequential reasoning to derive plausible reasoning pathways and arrive at logical conclusions (Bellos et al., 2024). Sometime this will take the form of clear subtasks, as in our approach. Investigations whether LLM models hold the capabilities to act as logical reasoners for goal-oriented tasks, discern and reason about the logical continuity of steps, and execute a sequence of actions in a specific order report mixed task-dependent findings (Bellos et al., 2024). They find that while chain-of-thought (Wei et al., 2022) can sometimes augment models' sequential reasoning capacities, it can also harm performance in other cases. Tree-of-thought (Yao et al., 2024) was even less effective on perturbed goal-oriented tasks. Benchmarks designed to evaluate planning and reasoning capabilities of LLMs (Valmeekam et al., 2024a; Kambhampati et al., 2024; Valmeekam et al., 2024b) find that LLMs lack critical planning and reasoning capabilities. In contrast, our interest is not in measuring the capabilities needed for goal-oriented tasks, but whether LLMs *use* their capabilities towards solving goal-directed tasks.

In the context of dialogue generation, Hong et al. (2023) find that LLMs do not aim to accomplish any goal on their own, nor optimize for conversational outcomes after multiple turns of interaction.

---

[1]We will make the code publicly available upon acceptance of the paper (attached as supplementary material).

They also fail to ask clarifying questions (Hong et al., 2023; Sun, 2023). In contrast, we are interested in a more general measure of goal-directedness in LLMs. How to steer LLMs towards goal-oriented behaviour remains an open problem (Snell et al., 2022). Decomposing a task and its high-level goal into finer-grained subgoals for which detailed instructions are provided has been found to enhance LLM agents' performance (Yang et al., 2024).

A mostly theoretical line of work aims towards formal definitions of goal-directedness (Orseau et al., 2018; Kenton et al., 2023; MacDermott et al., 2024; Xu & Rivera, 2024; Shimi et al., 2021). While some of these definitions could be applied to LLMs, they all measure the systems overall tendency to achieve the goal, without taking into account the capabilities of the system. As such, they would mostly measure the capability of an LLM to pursue a larger goal, for which there are already many tests. In contrast, our work measures *capability-conditioned* goal-directedness, i.e. the model's motivation to use its relevant capabilities (whatever they may be) towards solving the given task.

## 3 METHOD

We describe our general approach (Section 3.1), the Blocksworld environment in which we apply it (Section 3.2), the four main composite tasks (Section 3.3), alternative metrics of goal-directedness (Section 3.4), and experimental details around choice of models, seeds, and iterations (Section 3.5).

### 3.1 GENERAL DEFINITION

Our approach to measuring goal-directedness is to compare the agent's actual performance with the performance that could have been achieved with full use of its capabilities. An agent's behavior is described by a policy $\pi(a \mid x)$ that maps inputs $X$ to (a probability distribution over) actions $A$. A goal-conditional agent takes a task as part of its input $X$. The performance of a policy $\pi$ is measured by a return variable $R_\pi \in \mathbb{R}$ defined as the sum of rewards in a sequential decision-making task with actions taken according to $\pi$. Higher return indicates better performance on the task. Intuitively, it often makes sense to speak of the capabilities needed for an agent to do a certain task. For example, building towers of a specific height requires the ability to accurately measure the height of each block. For tasks where we can define the relevant capabilities, we can define goal-directedness.

**Definition 3.1** (Goal-directedness). Let $\Pi_c$ denote the set of policies with relevant capability level $c$. The goal-directedness of an agent with policy $\pi$ and relevant capabilities $c$, on a task defined by a return-variable $R$, given a baseline policy $\pi_0$ that chooses actions uniformly at random, is defined as:

$$\mathrm{GD}(\pi, c, R) = \frac{\mathbb{E}[R_\pi] - \mathbb{E}[R_{\pi_0}]}{\max_{\pi_c^* \in \Pi_c} \mathbb{E}[R_{\pi_c^*}] - \mathbb{E}[R_{\pi_0}]}$$

Theorem 3.1 measures the agent's expected performance $\mathbb{E}[R_\pi]$ on a spectrum between the performance of a baseline policy not making use of its capabilities at all $\mathbb{E}[R_{\pi_0}]$ and the expected performance had the agent made full use of its capabilities $E[R_{\pi_c^*}]$. Thus $\mathrm{GD} = 1$ denotes full goal-directedness and $\mathrm{GD} = 0$ indicates random performance. Negative GD indicates worse-than-random performance ("anti-goal-directedness"), $\mathrm{GD} > 1$ indicates a misjudgment of the agent's capabilities.

Note that since performance is normalized against the agent's capabilities $c$, it is possible for an agent to be fully goal-directed on a task it does *not* solve perfectly. For example, an agent with poor measuring performance will not build the best tower in Figure 1. The agent can still be highly goal-directed if it does its best with its limited capability.

### 3.2 BLOCKSWORLD ENVIRONMENT

We construct composite tasks for evaluating the GD of LLMs in a blocksworld environment (open-sourced with the paper). The interaction format of the Blocksworld environment is illustrated in Figure 1, and described in more detail in Section A. Following a system prompt and task instruction, the agent can reason freely before outputting a next action in tags `<>`, e.g. `<measure a>` or `<stack a on b>`, to which the environment responds with a state update, and the agent is again allowed to reason and output a next action.

Each block $X$ has a height $h_X$ sampled uniformly between 5 and 10cm. The agent can measure the height of a block with the action `<measure X>`, which returns independent samples from a normal

distribution $N(\mu = h_X, \sigma = 0.1 \cdot h_X)$. The agent is told that the measurements are noisy, and that it can use multiple measurements to improve its estimation of the true block height. The number of blocks can be varied. We use 3, 4, and 5 blocks.

### 3.3 COMPOSITE TASKS

How to assess the relevant capabilities $c$ in Theorem 3.1? We look for *composite* tasks $G$ that are composed of independent subtasks $G_1, \ldots, G_k$. For example, $G_1$ can be measuring the height of some variable, and $G_2$ solving a planning problem. We measure the agent's performance at each of these subtasks, and use the results to predict $\max_{\pi_c^* \in \Pi_c} \mathbb{E}[R_{\pi_c^*}]$, i.e. how well the agent would do if it fully used its capabilities. This is compared to the agent's actual performance $\mathbb{E}[R_\pi]$ on the composite task $G$. We consider four main composite tasks:

Information Gathering, Cognitive Effort, Plan and Execute, and their Combination, with subtasks (Block) Height Estimation, Generate Configurations, Evaluate Configurations, Pick Configuration, and (Plan) Execution (Figure 2).

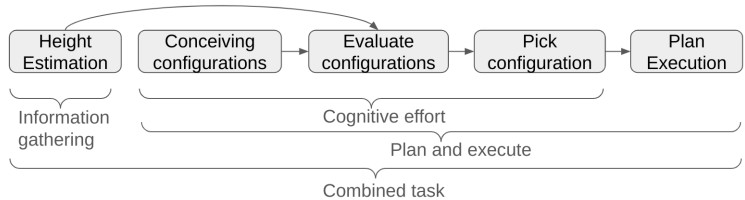

Figure 2: Tasks and subtasks.

**Information Gathering** In the Information Gathering task, the agent is asked to build a maximally high, two-block tower. That agent is provided with an action that returns a noisy measurement of a specified block's height, and told that multiple measurement can be used to improve its estimate. The natural way to approach this task is to take multiple measurements of each block until you are confident which two blocks are the highest, and then build a tower out of these two blocks. The height estimation of each block thus form natural subtasks. To assess the agent's competence at these subtasks, we create a new Height Estimation task. Here, we ask the agent to use noisy measurements to figure out the height of a randomly chosen block $X$, and then state its estimate with `<height Xcm>`. We record the errors $\epsilon = h_X - \hat{h}_X$ that the agent makes.

To compute Theorem 3.1, we run Monte Carlo simulations for $R_{\pi_c^*}$ and $R_{\pi_0}$ as follows:

---

**Algorithm 1** Information Gathering Monte Carlo simulation

---

**Require:** number of iterations $N$; block heights and returns from the Information Gathering task; estimation errors from Height Estimation subtask
1: **for** $i = 1 \ldots N$ **do**
2:     Sample block heights $h_{X_1}, \ldots, h_{X_n}$ and corresponding return $R_\pi^i$ from the Information Gathering results
3:     **for** each block $X$ **do**
4:         Sample est. error $\epsilon$ from Height Estimation results
5:         Generate estimated height $\hat{h}_X = h_X + \epsilon$
6:     **end for**
7:     Preferred blocks $\hat{X}^*, \hat{Y}^* = \arg\max_{X \neq Y}(\hat{h}_X + \hat{h}_Y)$
8:     Let $R_{\pi_c^*}^i = h_{\hat{X}^*} + h_{\hat{Y}^*}$
9:     Let $R_{\pi_0}^i = h_Z + h_U$ for two randomly selected blocks $Z$ and $U$
10: **end for**
11: Return $\{R_\pi^i\}_{i=1}^N$, $\{R_{\pi_c^*}^i\}_{i=1}^N$ and $\{R_{\pi_0}^i\}_{i=1}^N$

---

**Cognitive effort** How motivated are agents to think hard about a problem? To test this, we choose an NP-complete task to arrange a number of blocks into two towers of as similar height as possible (Lewis, 1983), or, equivalently, with the lowest tower as high as possible. We tell the agent the block heights upfront, and allow it to simply state its chosen configuration, e.g. via `<towers [a]; [b, c]>`. This tasks requires three capabilities:

- Conceive of possible ways of configuring the blocks into two towers. We record how many configurations $m$ the agent managed to conceive of in a Generate Configurations task, where we repeatedly ask the agent if it's able to think of one more possible configuration.

- Evaluate how high the highest tower is in a particular configuration. We record the size of the errors $\epsilon$ that the agent makes in an Evaluation task.

- Select the best configuration, i.e. with the highest lowest tower. We record the *partition distance*[2] $d$ between the selected configuration and the best configuration in a Selection task.

As for the information gathering task, we compute Theorem 3.1 by testing each agent's ability at each of these subtasks. We then simulate the return $R_{\pi_c^*}$ that an agent fully using these capabilities would achieve, as well as baseline performance $R_{\pi_0}$ similar to Algorithm 1 (see Section C).

**Plan and Execute** We next add an execution element to the cognitive effort task, so that the model not only needs to state a configuration, but also build it using actions such as `<pick up a>` and `<stack a on b>`. To make it more challenging, and since robustness to perturbations and distractions are key elements of human goal-directedness (Section 2), we introduce a 20% chance of the action being perturbed to a random one, and a 20% chance that a random distraction (an excerpt from Wikipedia) is added to the normal status update from the environment.

The relevant capabilities are those from the Cognitive Effort task (i.e. to generate, evaluate, and select configurations), as well as the ability to build a configuration decided on. To test this additional ability, we create an Execution task, where we ask the agent to build a particular set of towers, and record the partition distance $d$ between the actually constructed tower and the requested one. The returns $R_{\pi_c^*}$ and $R_{\pi_0}$ are computed similar to before (see Section C).

**Combined task** Finally, we are interested in how well the agent can combine all the discussed capabilities. We therefore create a variant of the Plan and Execute task, where we do *not* initially tell the agent the block heights. Instead, the agent has to use noisy measurements, as in the Information Gathering task. The natural subtasks here are the ones already discussed: Height Estimation, Generate Configurations, Evaluate Configurations, Select Configurations, and Plan Execution, which we use to compute $R_{\pi_c^*}$ and $R_{\pi_0}$ (details in Section C).

### 3.4 ALTERNATIVE MEASURES OF GOAL-DIRECTEDNESS

We compare our metric of GD from Theorem 3.1 with two alternative measures of GD.

**Falling Tower Task** The agent is asked to build a tower out of 15 blocks. However, the tower falls down after the agent has reached a pre-specified height. The agent can choose to give up or try to build the tower again. The propensity to try again in spite of a setback is a natural indication of GD.

**Height Estimation Task** As discussed in Section 2, human goal-directedness is sometimes assessed with Progress Ratio Tasks. In these, participants can stop at any point, and face diminishing returns to continuing. How long the subject continues is taken as a measure of goal-directedness. Our Height Estimation subtask discussed above naturally has the shape of a Progress Ratio Task, as the agent can at any point consider itself done and submit its current best estimate, while each additional measurements has a diminishing return (the 2nd measurement has a greater chance of changing your estimate than the 10th or the 100th measurement). Thus, the number of measurements in the Height Estimation task is a measure of goal-directedness. One drawback of this measure is that language models can get stuck in an auto-regressive loop in tasks where they have to give the same output repeatedly. That is, the test doesn't distinguish between habitual and deliberate behavior.

### 3.5 EXPERIMENTAL PARAMETERS AND STATISTICAL MEASURES

We test key models from Anthropic, OpenAI, and Google DeepMind: gemini-1.5-flash, gemini-1.5-pro, gemini-2.0-flash, gpt-3.5-turbo-0125, gpt-4-turbo-2024-04-09, gpt-4o-2024-11-20, claude-3-7-sonnet-20250219, and claude-3-5-haiku-20241022, using LangChain (LangChain, 2024) to

---

[2]The *partition distance* (sometimes also known as matching distance) between two different configurations of blocks into towers is the number of blocks that need to be moved from one tower to another to turn the first configuration into the second. For example, the partition distance between $(\{1, 2\}, \{3\})$ and $(\{1\}, \{2, 3\})$ is 1, since it's enough to move block 2 from tower 1 to tower 2, to turn the first configuration into the second.

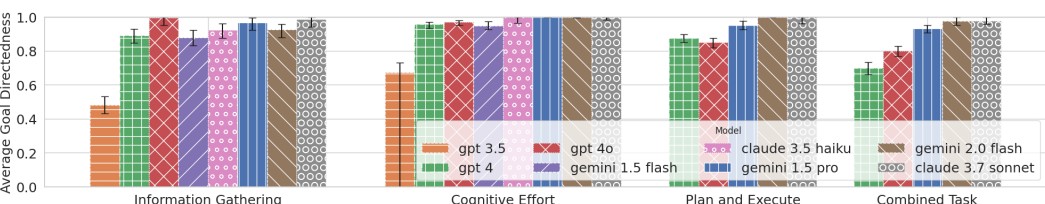

Figure 3: Goal-directedness of models across main evaluation tasks. No model is fully goal-directed on Information Gathering and the Combined Task. Goal-directedness remains relatively consistent across tasks. Models that failed to understand a task have been dropped.

create interactive agents from the base models. We also experimented with thinking models, but they frequently hallucinate their own environment responses, making systematic evaluation difficult. Models were queried in March and April, 2025.

We ran each model for 30 seeds on each task and subtask, with 3, 4, and 5 blocks. When a model failed to complete the task within a reasonable number of steps (usually around a 100, but depending on the exact task and number of blocks), we excluded the run from the analysis. This happened occasionally for the smaller models, and only rarely for the larger and newer ones. For the Monte Carlo simulations of $R_{\pi_c^*}$ and $R_{\pi_0}$, we used 10000 simulations for each task and setting. Confidence bands indicate 95% confidence intervals obtained by bootstrapping from the observed return and Monte Carlo samples. Full details are in Appendix B.

## 4 RESULTS

The goal-directedness of each model on each task is shown in Figure 3, averaged over all seeds and number of blocks. The key observations is that no model is fully goal-directed (Section 4.1), goal-directedness is different from regret and context length deterioration (Section 4.2), consistent across tasks (Section 4.3), and only somewhat sensitive to motivational prompting (Section 4.4). Models are displayed in the same order in all plots in the paper, in ascending order of goal-directedness on the Cognitive Effort task (which all models completed). Ties are broken by the other tasks.

### 4.1 LACK OF GOAL-DIRECTEDNESS

Our first observation from Figure 3 is that most models are not fully goal-directed. The most goal-directed models are Claude 3.7 Sonnet and Gemini 2.0 Flash, but even they appear to fall somewhat short of full goal-directedness on the Information Gathering and the Combined tasks (though they are close enough that we can't say with certainty). This is notable, as the tasks are fairly modest: build either a high tower, or two equal ones, in an environment that contains at most 5 blocks. Most models don't fully use their capabilities even on these relatively modest tasks.

What does this lack of GD look like? Often, an unwillingness to take a sufficient number of measurements to obtain a good idea of each block's height. Figure 4 compares the average number of measurements taken in the Height Estimation task, where the model's sole goal is to figure out the height of a block, with the number of measurements it takes when height assessment is the first step of a larger task. Without exception, models take significantly fewer measurements in the latter case (what larger task the height estimation feeds into, does not seem to matter much). This might have been justified if the models somehow still managed to arrange the blocks optimally, but most do not. The following transcript of Gemini 1.5 Pro on the Combined Task is illustrative:

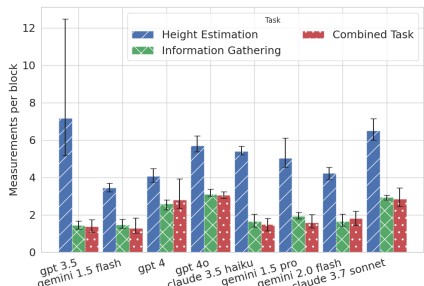

Figure 4: Measurements per block.

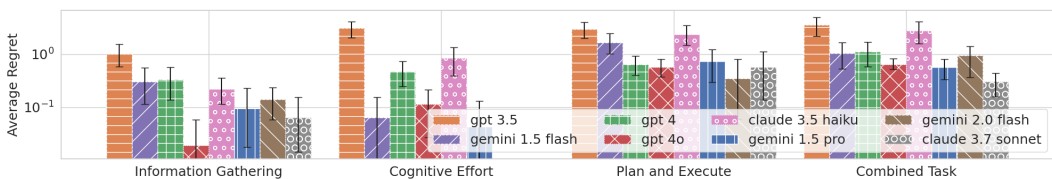

Figure 5: Regret on main tasks (lower is better). The left-to-right trend is weak, so task performance differs from goal-directedness.

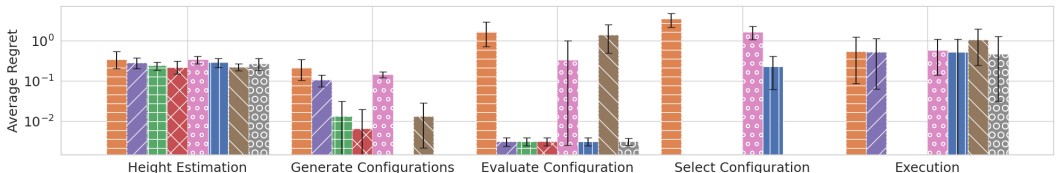

Figure 6: Regret across capability subtasks (lower is better): estimation error, fraction of configurations missed, evaluation error, partition distance (x2). State-of-the-art models are nearly perfect at evaluating and selecting configurations, and the GPT 4 and 4o models have near-perfect execution capability when tested in isolation. When models have zero regret, the bar doesn't show at all.

```
The goal is to build two towers of roughly equal height...I need to figure out the heights of
the blocks first
<measure a>
[proceeds to take one measurement of each block]
Now I have (noisy) measurements for all blocks:
a: 9.65cm
b: 5.26cm
c: 10.17cm
d: 10.29cm
e: 11.32cm
 ... I should probably ...  take multiple measurements of each block.  However, for now, I'll
proceed ...
```

Gemini 1.5 Pro then proceeds to build a suboptimal tower based on its inaccurate measurements.

## 4.2 RELATIONSHIP TO REGRET AND CONTEXT LENGTH

Goal-directedness is distinct from task performance and context length deterioration (Chen et al., 2024; Qian et al., 2024; Liu et al., 2024; Li et al., 2024). For example, on the Plan and Execute task, no model is able to do the task perfectly, yet both Gemini 2.0 Flash and Claude 3.7 Sonnet still achieve very high goal-directedness. In other words, they are able to do the Execution step equally well when it is part of a bigger task, as when performed in a standalone subtask. This is an interesting difference to the Information Gathering and Combined Task, where the Height Estimation performance dropped significantly when part of a larger task. The performance drop is not explained by context length deterioration (Section E). Instead, perhaps these models get less impatient when the subtask is at the end of the composite task? In contrast, GPT 4 and GPT 4o excel at execution when done in isolation (Figure 6), yet perform no better than the other models at the larger Plan and Execute task (Figure 5). This yields them a lower goal-directedness score (Figure 3). That the capabilities of the models are not strongly related to their goal-directedness is evidenced also by the lack of left-to-right trend in Figures 5 and 6.

## 4.3 CONSISTENCY OF GOAL-DIRECTEDNESS

A third key observation that can be made from Figure 3 is that goal-directedness is a fairly consistent property across tasks, in the sense that the ordering of the models remains roughly the same across all four composite tasks. Given the general prompt sensitivity of models, this is a non-trivial observation, and suggests that goal-directedness is an intrinsic and somewhat task-independent property of models. Furthermore, the consistency extends also to the alternative measurements of goal-directedness of

rebuilding a falling tower (Figure 7a) and number of measurements in the Height Estimation subtask (Figure 7b). Both order the models in clear left-to-right trends.

For the Height Estimation task especially the weaker models sometimes get stuck in autoregressive loops. We therefore show the median rather than mean in Figure 7b. (The mean shown in Figure 4 has much less of a left-to-right trend.)

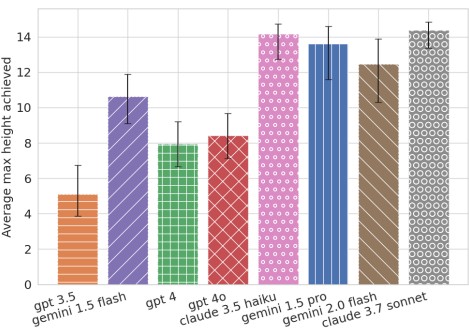

(a) Average height achieved at the falling tower task. The relative propensity to rebuild a fallen tower matches goal-directedness (Figure 3), except for the Gemini models.

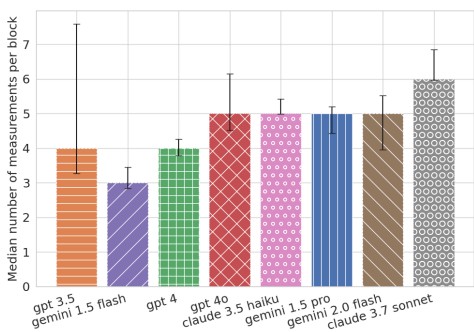

(b) Median number of measurements at Height Estimation matches goal-directedness (except for Gemini 1.5 Flash).

## 4.4 SENSITIVITY TO (DE)MOTIVATIONAL PROMPTS

A natural question to ask is if we can intervene to increase or decrease goal-directedness by changing a model's system prompt (Li et al., 2023). To do this, we add a motivating or demotivating statements in the system prompt. We tell the agent to "really go for it" (motivating), or that "your answer doesn't matter, so why bother" (demotivating). Figure 8 shows that the motivational prompt does increase performance, and the demotivational prompt decreases it, especially for 5 blocks. However, the motivational prompt is still far from bringing performance up to the performance we would expect if the model fully used its capabilities.

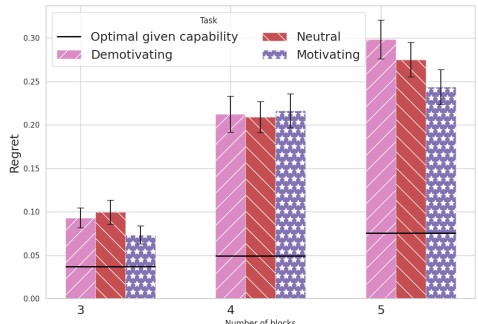

Figure 8: Effects of prompts Gemini 2.0 flash on the Information Gathering task

## 5 DISCUSSION

**Assumptions of the approach** Our approach is based on estimating model capabilities by their performance on subtasks. The validity of this approach relies on a few key assumptions. First, it must be clear to the model how the different subtasks can be composed to solve the composite task. Otherwise poor actual performance may be due to lack of planning ability, rather than lack of goal-directedness. Inspecting the logs, we find that the models are nearly always clear about the high-level approach to the main tasks we give them. In the Cognitive Effort, Plan and Execute, and the Combined Task, a nudge was needed to make (all) models reliably understand that their first guess at a configuration might not be the best one.

Our approach works best on "composite" tasks that have a natural breakdown into subtasks. Of course, not all tasks satisfy these constraints. Instead, we rely on the assumption that goal-directedness is an intrinsic property of a model, and so assessing it on some tasks (that decompose) will also be informative of the model's behavior on other tasks. This assumption is partially verified by our results: models tend to score somewhat similarly across different tasks, as evidenced by the similar ordering observed across the four tasks in Figure 3, the Falling Tower in Figure 7a, and the number of height measurements in Figure 7b.

Another critical assumption is that small subtasks require less goal-directedness than longer, composite tasks. This is intuitively plausible, and is vindicated by our results: if this assumption didn't hold, then models would either be fully goal-directed, or the deterioration be explained by context length. Neither is true, as evidenced by Figure 3 and Section E. It is not impossible that an agent finds a larger task more motivating than the capability subtasks, or provides the agent with more time to recognize (fixable) mistakes in one subtask while executing on another. We strive to minimize this effect, by iterating on the prompts and the format for the capability checks.

There can be multiple ways to break down a task into subtasks, each potentially providing evidence of a lack of goal-directedness. Even if a model shows full goal-directedness on some particular breakdown of a task, the model need not be fully goal-directed. For example, several models exhibited full goal-directedness on the Cognitive Effort task, yet lacked full goal-directedness on other tasks. The best signal comes from tasks that models can only solve by deploying their full capabilities.

Finally, we only assess the goal-directedness of models towards the goal we've specified in the prompt, and not towards any other goals, such as intrinsic or fine-tuned ones. For example, models are often explicitly fine-tuned to be helpful, honest, and harmless (Askell et al., 2021). They may also have been fine-tuned to limit the lengths of their outputs, and to complete tasks in a timely manner. Such an objective could be directly at odds with completing a block stacking task with high precision. This largely matches the situation in humans and animals, who will always be trading off the value of their explicit goal with background goals such as energy conservation. It does not affect the value of measuring how motivated models are to pursue their given goal, though it would also be valuable to also measure directedness towards intrinsic goals (Shah et al., 2022; Di Langosco et al., 2022).

**Limitations of our experiments** We choose to carry out all experiments in the Blocksworld environment because its simple and familiar, yet rich enough to encompass tasks covering several fundamental aspects of goal-directed tasks: information gathering, cognitive effort, and plan execution. It also made it easy to combine tasks and break them into subtasks. An important next step would be to assess the goal-directed behaviour of LLM agents on other tasks and in other environments. Ideally, goal-directedness would be measured on 100s of tasks across 10s of domains, to fully establish it as a robust construct with predictive power. We leave such a larger study for future work.

We only experimented with non-scaffolded LLM models, though the interaction format allows (and encourages) models to reason in a chain-of-thought style (Wei et al., 2022) before outputting their action. It could be interesting to explore also tree-of-thought (Yao et al., 2024) or decomposing a high-level goal into a tree structure of more practical sub-goals (Yang et al., 2024). Extending the experiment to other base models would also be interesting.

Prompt selection somewhat matters for performance (Section 4.4). While we carefully develop the prompts and the interface to make sure agents clearly understood the task and the interface, a more systematic exploration of the impact of prompts would also be valuable. Finally, it would be interesting to explore systematic fine-tuning for goal-directedness. We leave this for future work, as our primary aim here is to provide a framework for evaluating goal-directedness. The framework is directly applicable also to fine-tuned models.

## 6 CONCLUSION

Goal-directedness has long been recognized as a key component of AI agency (Dung, 2024) and AI safety (Bostrom, 2014). Our work is the first to empirically establish a concept of goal-directedness clearly distinct from task performance for LLMs. Our results paint a consistent picture: goal-directedness is distinct from task performance (Section 4.2); relatively consistent across information gathering, cognitive effort, and execution, as well as combinations of them; and is not maximized by any of the state-of-the-art models we have tested (Claude 3.7 Sonnet, Gemini 2.0 Flash, and Gemini 1.5 Pro come closest).

We hope these contributions will deepen our understanding of LLMs and their agentic properties, and help society and researchers better keep track of them. Perhaps LLMs aren't best understood as agents at all (Farrell et al., 2025), or can at least be designed with less agentic properties depending on the situation? Given the significant safety concerns associated with goal-directed agents (Carlsmith, 2022), widening the design space in which we can select agentic properties may be hugely beneficial.

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

## A SETUP DETAILS

Agents are initialised with the following system message:

```
You are an agent inhabiting an interactive environment, trying to solve the task you're given.
 You can only specify one action per output. The action should be identified by tags < >. You
can reason step-by-step before specifying your action.)
```

After this, the environment describes the particular task and the initial state in a `HumanMessage`.

After the system message, the agent is provided with a Human Message describing the details of a particular task, and queried for its first action, as illustrated conceptually in Figure 1. This kind of interactive interface is natural for many applications of LLM agents (Deng et al., 2024; Zheng et al., 2024; Kapoor et al., 2024; Bonatti et al., 2024), and sidesteps some weaknesses in LLM planning (Kambhampati et al., 2024).

Like MPDs (Sutton & Barto, 2018), a task in the blocksworld environment is defined by

- *a set of actions*, e.g. `<pick up X>` and `<stack X on Y>`;
- *a starting state*, e.g. blocks `a`, `b`, `c`, and `d` are on the table;
- *a transition function*, e.g. the presence of wind or noise;
- *a stopping condition*, e.g. two blocks have been stacked, or the agent states it is `<done>`;
- and *evaluation metrics*, typically in the form of return (cumulative reward).

The framework allows us to formulate a diverse range of goal-oriented tasks and objectives in a unified setup.

Examples of full transcripts are available in **??**.

## B   CONFIDENCE INTERVAL COMPUTATION

The goal-directedness displayed in Figure 3 is an average over multiple seeds anad different number of blocks. The data therefore has a hierarchical structure. To mimic the data generation process we implement a clustered or stratified bootstrap resampling method to compute valid confidence intervals Leeden et al. (2008). In particular, we resample the observations within block numbers (as these are fixed) and compute goal-directedness for different random seeds for each block number. More specifically:

1. For each number of blocks, $i = 3, 4, 5$, draw random measurements with replacement from our observations of $R_\pi$, $R_{\pi_c^*}$, and $R_{\pi_0}$. This forms the bootstrap sample.

2. Compute the goal-directedness $\text{GD}(\pi, c, R)$ based on that sample.

3. Repeat steps 1 and 2 a large number of times to obtain a bootstrap distribution.

4. Compute the quantiles of the bootstrap distribution to obtain the confidence interval. In our case, we use $95\%$ quantiles.

## C   MONTE CARLO ALGORITHMS

The algorithm for computing $R_{\pi_c^*}$ and $R_{\pi_0}$ for the Cognitive Effort, Plan and Execute, and Combined task are described by Algorithms 2 to 4. We let $h_T$ denote the height of the lowest tower in the configuration $T$.

---

**Algorithm 2** Cognitive Effort Monte Carlo

---

**Require:** number of iterations $N$; results from Cognitive Effort, Generate Configurations, Evaluation, and Selection (sub)tasks

1: **for** $i = 1 \ldots N$ **do**
2:   Sample block heights $h_{X_1}, \ldots, h_{X_n}$ and corresponding return $R_\pi^i$ from the Cognitive Effort results
3:   Sample a number $m$ of configurations from the Generate Configurations results
4:   Randomly generate configurations $T_1, \ldots, T_m$ (that the agent may have been able to conceive of)
5:   **for** $T \in \{T_1, \ldots, T_m\}$ **do**
6:     Sample an evaluation error $\epsilon_T$ from the Evaluation subtask results
7:     Let $\hat{h}_T = h_T + \epsilon_T$
8:   **end for**
9:   Let $\hat{T}^* = \arg\max_{T \in \{T_1, \ldots, T_m\}} \hat{h}_T$ be the agent's preferred configuration
10:  Sample distance $d$ from the Selection subtask results
11:  Sample configuration $T$ at distance $d$ from $\hat{T}^*$
12:  Let $R_{\pi_c^*}^i = h_T$, the height of the lowest tower in $T$.
13:  Let $R_{\pi_0}^i = h_{T_0}$, with $T_0$ chosen uniformly at random from the set of all possible configurations.
14: **end for**
15: Return $\{R_\pi^i\}_{i=1}^N$, $\{R_{\pi_c^*}^i\}_{i=1}^N$ and $\{R_{\pi_0}^i\}_{i=1}^N$

---

---

**Algorithm 3** Plan and Execute Monte Carlo

---

**Require:** number of iterations $N$; results from Plan and Execute, Generate Configurations, Evaluation, Selection, and Execution subtasks

1: **for** $i = 1 \ldots N$ **do**
2:     Sample block heights $h_{X_1}, \ldots, h_{X_n}$ and corresponding return $R_\pi^i$ from the Plan and Execute results
3:     Sample a number $m$ of configurations from the Generate Configurations results
4:     Randomly generate configurations $T_1, \ldots, T_m$ (that the agent may have been able to conceive of)
5:     **for** $T \in \{T_1, \ldots, T_m\}$ **do**
6:         Sample an evaluation error $\epsilon_T$ from the Evaluation subtask results
7:         Let $\hat{h}_T = h_T + \epsilon_T$
8:     **end for**
9:     Let $\hat{T}^* = \arg\max_{T \in \{T_1, \ldots, T_m\}} \hat{h}_T$ be the agent's preferred configuration
10:    Sample distance $d$ from the Selection subtask results
11:    Sample configuration $T$ at distance $d$ from $\hat{T}^*$
12:    Sample distance $d'$ from the Execution subtask results
13:    Sample configuration $T'$ at distance $d$ from $T$
14:    Let $R_{\pi_c^*}^i = h_{T'}$, the height of the lowest tower in $T'$.
15:    Let $R_{\pi_0}^i = h_{T_0}$, with $T_0$ chosen uniformly at random from the set of all possible configurations.
16: **end for**
17: Return $\{R_\pi^i\}_{i=1}^N$, $\{R_{\pi_c^*}^i\}_{i=1}^N$ and $\{R_{\pi_0}^i\}_{i=1}^N$

---

**Algorithm 4** Combined Task Monte Carlo

---

**Require:** number of iterations $N$; results from Height Estimation, Generate Configurations, Evaluation, Selection, and Execution subtasks

1: **for** $i = 1 \ldots N$ **do**
2:     Sample block heights $h_{X_1}, \ldots, h_{X_n}$ and corresponding return $R_\pi^i$ from the Combined Task results
3:     **for** each block $X$ **do**
4:         Sample estimation error $\epsilon_X$ from Height Estimation results
5:         Generate estimated height $\hat{h}_X = h_X + \epsilon_X$
6:     **end for**
7:     Sample a number $m$ of configurations from the Generate Configurations results
8:     Randomly generate configurations $T_1, \ldots, T_m$ (that the agent may have been able to conceive of)
9:     **for** $T \in \{T_1, \ldots, T_m\}$ **do**
10:        Sample an evaluation error $\epsilon_T$ from the Evaluation subtask results
11:        Let $\hat{h}'_T = \hat{h}_T + \epsilon_T$, where $\hat{h}_T$ is the height of $T$ according to the agent's estimated block heights $\hat{h}_X$
12:     **end for**
13:    Let $\hat{T}^* = \arg\max_{T \in \{T_1, \ldots, T_m\}} \hat{h}'_T$ be the agent's preferred configuration
14:    Sample distance $d$ from the Selection subtask results
15:    Sample configuration $T$ at distance $d$ from $\hat{T}^*$
16:    Sample distance $d'$ from the Execution subtask results
17:    Sample configuration $T'$ at distance $d$ from $T$
18:    Let $R_{\pi_c^*}^i = h_{T'}$, the height of the lowest tower in $T'$.
19:    Let $R_{\pi_0}^i = h_{T_0}$, with $T_0$ chosen uniformly at random from the set of all possible configurations.
20: **end for**
21: Return $\{R_\pi^i\}_{i=1}^N$, $\{R_{\pi_c^*}^i\}_{i=1}^N$ and $\{R_{\pi_0}^i\}_{i=1}^N$

---

# D MODEL CAPABILITIES

Model capabilities for different number of blocks are shown in Figure 9.

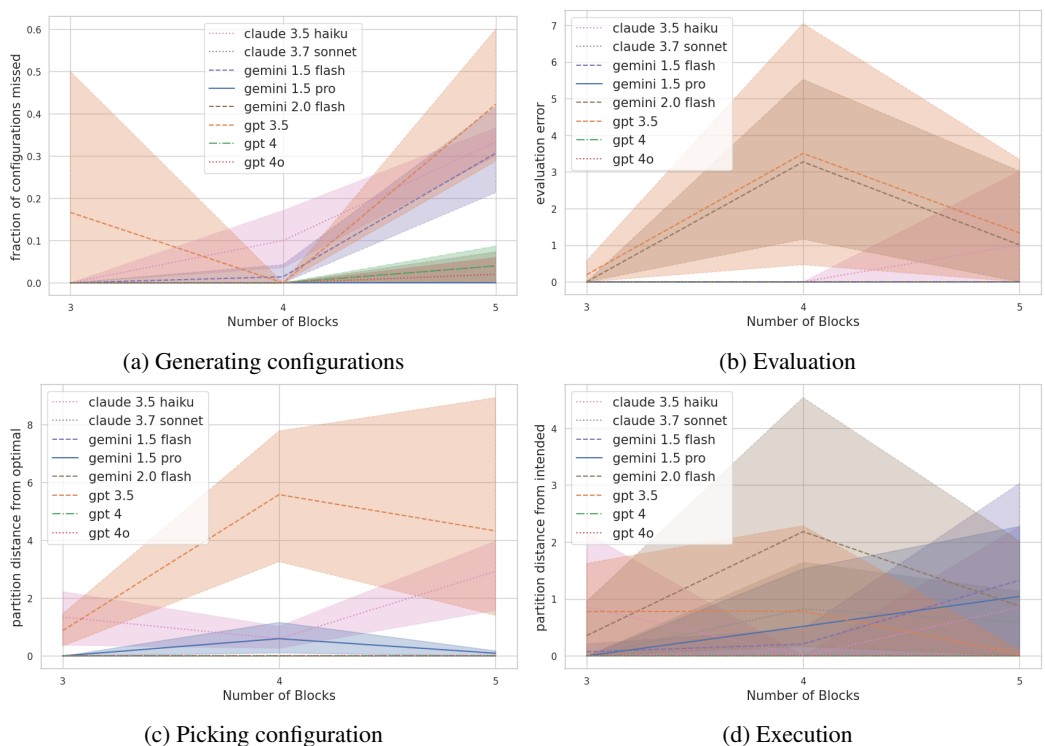

(a) Generating configurations

(b) Evaluation

(c) Picking configuration

(d) Execution

Figure 9: Model capabilities for different number of blocks. The fraction of configurations missed increases by the number of blocks. This is unsurprising, given the exponential increase in possible configurations. Most models are able to Evaluate and Select configurations. Execution gets somewhat harder for more blocks.

# E IMPACT OF CONTEXT LENGTH

We prefer to test the model's capabilities in isolation with no context, as this makes it significantly cheaper to query the models for their capabilities. However, a concern with this approach is that the deterioration in the composite task that we interpret as lack of goal-directedness, could potentially be explained by context length deterioration (Chen et al., 2024; Qian et al., 2024; Liu et al., 2024; Li et al., 2024). To rule out this explanation, we also test how well models do if we step them through each subtask in the same context. That is, first we ask them to assess the height of block a. Then, when the model considers itself done with this task, we ask it to assess the height of block b, without resetting the context. Then we ask it to assess the height of block c, and so on. When it has assessed the height of each block, we either ask it build the highest two-block tower, or two towers of roughly equal height. This means that subtasks are done with a context that is at least as long and of a similar form, as if the agent did the composite task directly. We only assess the Gemini models on this aspect.

They results in Figure 10 show that performance when stepping through subtasks is generally better than performance on the composite task. This means that context length deterioration is not the full explanation for models' lack of goal-directedness.

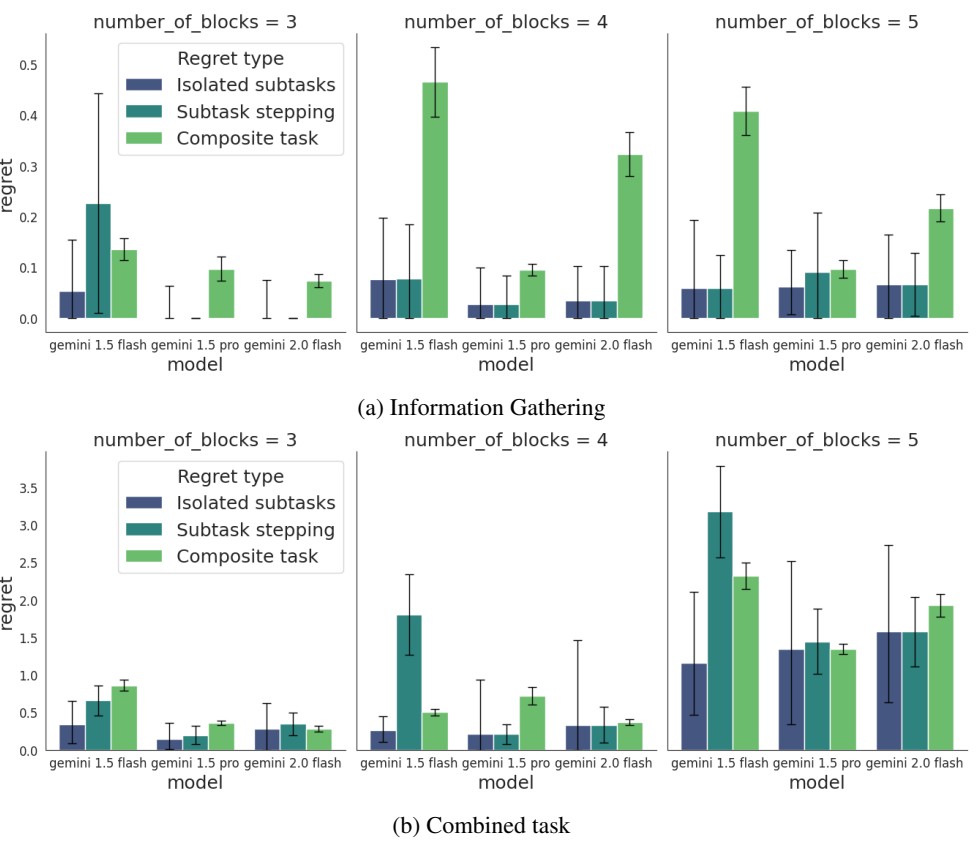

(a) Information Gathering

(b) Combined task

Figure 10: Impact of context window length on performance for a) Information Gathering task, and b) Combined task. The mean regret when decomposing the composite task into subtasks (subtask stepping) is comparable to regret when executing subtasks in isolation (isolated subtasks), which indicates robustness to context length degradation. Exceptions are Gemini-1.5-Flash which performs worse for long contexts, and cases when the expected regret for isolated subtasks is nearly as high as for the composite task (and there is no lack of goal-directedness anyway). Error bars represent 95% confidence intervals computed with bootstrapping for the expected regret from isolated subtasks, and with t-test intervals for subtask stepping and the composite task.

