# OpenReview forum: "Evaluating LLM Goal-Directedness"
_ICLR.cc/2026/Conference — Submitted to ICLR 2026_

### Official Review · Reviewer_wBwk · 2025-10-28

**Soundness:** 2
**Presentation:** 2
**Contribution:** 3
**Rating:** 4
**Confidence:** 3

**Summary:**

The paper aims to define and evaluate "goal-directedness" (GD) in LLMs. This is different from task performance, which conflates capability (is the model capable of completing the goal) and GD (how strongly does the model *try* to complete the goal). The key question is how to measure GD in a way that is independent from capabilities. To do so, the authors design text-based "Blocksworld" tasks which naturally decompose into subtasks. For example, one task is to stack three blocks into two towers of similar height. The model has access to a "stack" action and a "measure" action, which returns noisy measurements. The natural way to complete this task is to (1) take many measurements of each block's height, (2) decide on the optimal configuration, and (3) make that configuration. Each of these tasks can also be tested in isolation, e.g., "determine the height of block A". If a model performs well on each subtask in isolation, one can conclude that it has the necessary capabilities for the overall task. If such a model performs poorly on the overall task despite this, the authors argue that this indicates poor GD. The authors evaluate 8 proprietary LLMs in this way and find that most models are fairly goal-directed but not fully goal-directed, and GD is relatively consistent across the tasks they consider. The authors also show that (de)motivational prompting (e.g., "really go for it" or "this doesn't really matter") has nonzero but limited impact on GD.

**Strengths:**

GD in LLMs is clearly an important topic. I think the core idea of the paper -- separating capabilities from GD by comparing subtask performance and task performance -- is quite interesting and creative. While this idea has limitations, I think it's a pretty good first attempt. The experiments show that the idea works reasonably well, at least in this setting. I particularly appreciated the frank discussion of limitations by the authors.

**Weaknesses:**

I have some interrelated writing concerns and methodology concerns. I don't think any of these concerns are dealbreakers individually, and for an initial proof-of-concept paper like this, I don't expect completely airtight experiments. However, it does seem like there are a lot of things that could be cleaned up. Overall, I'm inclined towards acceptance *if* the issues below can be addressed. My current rating of weak reject is based on the current state of the paper (i.e., that would be my rating if no changes are made).

**Concern 1**: The task design and the subtask decompositions seem somewhat arbitrary. I also initially found the descriptions quite confusing.

1a. I initially was very confused about what the tasks were (Section 3.3). My understanding now is that there are four main overall tasks: Information Gathering (IG), Cognitive Effort (CE), Plan and Execute (PE), and their Combination. IG only has one subtask, which is height estimation. CE and PE have mostly overlapping subtasks, including one which confusingly is called "plan execution". Also, I initially thought IG, CE, and PE were subtasks of the main overall task. Can you confirm that my understanding is correct?

1b. IG is listed as having one subtask (height estimation). But even after the heights are estimated, it still has to select to two tallest blocks and stack them. Why is that not also a subtask? And if a model is good at height estimation but bad at IG, how do you know that shows low GD and not that it's bad at stacking?

1c. GD is conditioned on a capability level c, and Definition 3.1 requires optimizing over the set of policies with that capability level. My understanding is that you do this by using a Monte Carlo algorithm which takes the subtask outputs as inputs, i.e., the optimal policy given the subtask outputs. But it seems to me that this only works if the overall task fully decomposes into subtasks, which I already argued is not true for IG. It's harder for me to tell whether it's true for the other tasks.

1d. How is regret defined? I understand that 0 is best, but it's hard to interpret Figures 5 and 6 without knowing the scale of regret. For example, what does regret = 10 indicate? Is that good or bad?

**Concern 2**: Some of the methodological choices seem concerningly ad-hoc, specifically with regards to excluding certain data.

2a. "When a model failed to complete the task within a reasonable number of steps (usually around a 100, but depending on the exact task and number of blocks), we excluded the run from the analysis." Why are failed runs excluded rather than considered failures? Also, how did you choose the max number of steps? Do the results change if this maximum is varied? What percentage of runs is this relevant to?

2b. "Models that failed to understand a task have been dropped." Similarly, why are failed runs excluded rather than considered failures? Also, how did you determine whether a model "failed to understand" the task?

2c. "We also experimented with thinking models, but they frequently hallucinate their own environment responses, making systematic evaluation difficult." If thinking models are truly so bad at these tasks that GD cannot even be evaluated, that is quite surprising, since thinking models are supposed to be better at multi-step tasks like these. It's especially surprising that this holds for _all_ of the thinking models the authors tested (although the paper doesn't mention how many thinking models were tested). I think this deserves more exploration. Assuming the authors are correct, I don't think it's their responsibility to fully debug these thinking models, but I think they should rigorously confirm that their evaluation is correct (and provide evidence in the paper to support that).

**Concern 3**: These tasks seem overall fairly easy. If future models solve both the subtasks and the overall tasks perfectly, does that imply that those models are 100% goal-directed? Or are the tasks just so easy that it doesn't require much GD to solve them? The authors state that GD was fairly consistent across tasks in their experiments, but maybe that was just because all of the tasks are quite similar in structure and difficulty level. The authors do acknowledge that their restricted Blocksworld environment has limitations, which I appreciate, although they don't mention this specific limitation. I don't think the authors need to solve this problem in order for the paper to be publishable, but it would be nice to include a discussion of this in the paper.

**Minor feedback**. Some of these are phrased as questions, but I don't expect responses. These points are all minor enough that none of them are affecting my overall evaluation of the paper.
1. "But task performance arguably says more about capability than goal-directedness". Why? It seems to me that task performance is basically the combination of capability with goal-directedness.
2. Definition 3.1: I think "capability" should probably be defined before using that term in a definition.
3. Definition 3.1: What about division by zero? This can happen if e.g. a random policy is optimal, right?
4. You might consider a different name than "Blocksworld", which is quite similar to "Gridworld", which is quite different.
5. I think "Theorem 3.1" should be "Definition 3.1" everywhere.

**Questions:**

I mixed in my questions with the weaknesses above. My top priority is making sure that I fully understand the experiment design and the rationale behind it. Once I'm confident that I understand everything, I can make more concrete recommendations for revisions and what specific factors would influence my final rating.

---

> ### Author Response · Authors · 2025-11-26
>
> Thank you for your review of our paper and for highlighting the strengths of our work, that separating capabilities from GD by comparing subtask performance and task performance is interesting and creative. We would like to address your concerns:
>
> Concern 1: a) Yes, your understanding of evaluation tasks is correct. There are four main composite tasks: Information Gathering (IG), Cognitive Effort (CE), Plan and Execute (PE), and Combined. These are the tasks on which we measure goal-directedness. The subtasks (Height Estimation, Generate Configurations, Evaluate Configurations, Select Configuration, Execution) are used to infer model capabilities. IG, CE, and PE can also be viewed as both standalone tasks, but also as subtasks of the main Combined task.
>
>
> b) This is a great observation! You are right that after height estimation, the model must also select the two tallest blocks and stack them. We did not include these as separate subtasks because:
> Selection is trivial given accurate estimates: Once heights are known, selecting the two tallest is a simple argmax operation. In our experiments, models virtually never err on this step when given accurate height information, errors in final tower height trace back to measurement errors, not selection errors.
>
>
> Stacking is deterministic in IG: Unlike in Plan and Execute (where perturbations and distractions create execution challenges), the Information Gathering task has no stochastic execution—stacking succeeds deterministically. Thus there is no capability to measure.
> We can verify this empirically: if a model is good at height estimation but bad at IG, we can inspect whether the failure occurs at selection or stacking. In practice, we do not observe this pattern - failures trace to the model taking insufficient measurements.
> We will add this clarification to the paper and include a brief analysis confirming that selection/stacking errors are negligible in IG.
> c) Definition 3.1 requires knowing the optimal policy given capability level c, which we approximate via Monte Carlo simulation over subtask performances.
> For IG, our Monte Carlo simulation (Algorithm 1) does implicitly include selection: given estimated heights h^_X for each block, line 7 computes the preferred blocks as argmax, and line 8 computes the return based on the true heights of those selected blocks. Thus selection is modeled, just not as a separately measured subtask (because, as noted above, selection given estimates is trivial and error-free in practice).
> We acknowledge that if subtasks interact in complex ways not captured by our decomposition, the Monte Carlo estimate of \pi*_c could be biased. We will add discussion of this assumption and its limitations. For the tasks studied here, we believe the decomposition is appropriate, but we agree this is an important consideration for extending the framework to more complex environments.
>
>
> d) Regret is defined as the difference between optimal performance and actual performance:
> Regret = E[R_π*] − E[R_π]
> where R_π* is the return of the optimal policy (with perfect capabilities) and R_π is the model's actual return.
> For our tasks, regret is measured as follows:
> In Information Gathering: regret = (height of optimal two-block tower) − (height of tower actually built)
> In Cognitive Effort / Plan and Execute / Combined: regret = (height of lowest tower in optimal configuration) − (height of lowest tower actually built)
> So regret = 10 cm means the model's tower was 10cm shorter than optimal. Given block heights range from 5–10cm, regret of 10 represents a substantial error (e.g., choosing two 5cm blocks instead of two 10cm blocks).
> We will add this definition explicitly to the paper and include guidance on interpreting the scale in Figures 5 and 6.
>
> Concern 2:
> a) We excluded runs exceeding the step limit because these typically represent degenerate behaviors (e.g., infinite loops, repeated invalid actions) rather than meaningful task attempts. Including them as "failures" with minimal return would conflate inability to engage with the task interface with lack of goal-directedness.
> To address your concerns, we will report the percentage of excluded runs per model (this was rare for larger models—typically <5%—and more common for smaller models).

---

> > ### Author Response · Authors · 2025-11-26
> >
> > b) This occurred specifically for the Cognitive Effort, Plan and Execute, and Combined tasks, where some models initially did not understand that their first configuration guess might not be optimal. As noted in Section 5: "a nudge was needed to make (all) models reliably understand that their first guess at a configuration might not be the best one."
> > We determined this by inspecting outputs: models would propose a single configuration and immediately declare completion without exploring alternatives, even when the prompt asked them to find the best configuration. After adding clarifying instructions, all models engaged appropriately with the task.
> > We will clarify in the paper that this involved prompt iteration, not post-hoc data exclusion.
> >
> > c) Thinking models hallucinating environment responses:
> > We tested o1 and o1-mini. The issue is that these models, during their extended reasoning, would sometimes generate hypothetical environment responses and then act on those hallucinated responses rather than waiting for actual environment feedback. This is a known issue with thinking models in agentic settings.
> > We agree this deserves more exploration. We will add example transcripts showing the hallucination behavior. We also note this as important future work as thinking models improve.
> >
> > Concern 3:
> >
> > Although the tasks overall seem easy, current models cannot solve both subtasks and composite tasks perfectly (this would indicate GD = 1), meaning despite the simplicity of our conceptual framework current LLMs  are still not fully goal-directed on these tasks.
> >
> > Thank you for your comments and suggestions, we will incorporate them in the updated draft of our paper!

---

### Official Review · Reviewer_qUGp · 2025-10-31

**Soundness:** 3
**Presentation:** 3
**Contribution:** 2
**Rating:** 4
**Confidence:** 2

**Summary:**

This paper analyzes the differences in LLMs' performances on sub-tasks and the combined tasks, denoted as goal-directness. The metric is defined as $(R_\pi-R_\text{random})/(R_\max-R_\text{random})$ where $R_\max$ is infered from sub-tasks' performances. It evaluates LLMs' goal directness in blocksworld where the model can measure the heights of blocks (with noise) and stack blocks to achieve tasks (such as to build two towers of similar heights). It mainly evaluates older LLMs such as gpt-3.5, 4o, and claude-3-7-sonnet and analyzes their performances.

**Strengths:**

It's interesting to evaluate the goal-directness of LLMs.

The paper is generally well-written.

**Weaknesses:**

* The blocksworld environment is synthetic and simple. It would be better to evaluate the models' performance in more realistic settings such as web/code agents.
* The evaluated LLMs are old. The results may not be informative enough to guide current actions.
* There are other factors to explain performance differences on combined tasks other than the goal-directness. For example, some models may prefer shorter outputs / fewer iterations than other models, which would make the models choose to stop earlier in height estimations and thus get worse performances.

**Questions:**

* Are there results on more realistic tasks and more recent LLMs?
* How to distinguish other factors to explain the performance differences on combined tasks?

---

> ### Author Response · Authors · 2025-11-26
>
> Thank you for your review of our paper! We would like to address your concerns:
>
>  We acknowledge that Blocksworld is synthetic and simple. However, our choice of this environment was deliberate – a controlled environment allows precise measurement of capabilities via subtask decomposition, interpretable failure modes, and clean separation of the factors we aim to study. As discussed in Section 5, establishing goal-directedness as a robust construct requires first demonstrating it in settings where confounds can be minimized.
>
> That said, we agree that generalization to realistic settings is important. We note several reasons to expect our findings would transfer:
> Consistency across diverse subtask types: Our four main tasks span information gathering, cognitive effort, and plan execution - these are fundamental components of web/code agent behavior. The consistent ordering of models across these diverse tasks (Figure 3) suggests goal-directedness is an intrinsic model property rather than an artifact of Blocksworld specifics.
>
>
> The framework is environment-agnostic: Definition 3.1 applies wherever tasks decompose into capability-revealing subtasks. Web agents (e.g., Mind2Web, WebArena) and code agents (e.g., SWE-bench) naturally decompose into navigation/retrieval subtasks, reasoning subtasks, and execution subtasks—directly analogous to our structure.
>
>
> Existing evidence from adjacent work: Hong et al. (2023) [1], Chinmaya et al [2] report that LLMs fail to optimize for conversational outcomes or ask clarifying questions in dialogue settings – a naturalistic finding consistent with our measured lack of goal-directedness in information gathering.
> We would be happy to add discussion of how the framework extends to realistic settings, and to explicitly frame broader evaluation as important future work. We believe the controlled Blocksworld setting is valuable precisely because it isolates goal-directedness from the many confounds present in complex realistic environments.
>
> [1] Hong, Joey, Sergey Levine, and Anca Dragan. "Zero-shot goal-directed dialogue via rl on imagined conversations." arXiv preprint arXiv:2311.05584 (2023).
>
> [2] Andukuri, Chinmaya, Jan-Philipp Fränken, Tobias Gerstenberg, and Noah D. Goodman. "Star-gate: Teaching language models to ask clarifying questions." arXiv preprint arXiv:2403.19154 (2024).
>
> Model Recency
> We note that our evaluated models (queried March–April 2025) include recent releases: Claude 3.7 Sonnet (released February 2025), Gemini 2.0 Flash, and GPT-4o-2024-11-20. These represent state-of-the-art at the time of evaluation.
> More importantly, we view the paper's primary contribution as establishing a framework for the evaluation of goal-directedness, not as benchmarking specific model versions. Our key findings that goal-directedness is distinct from capability, consistent across tasks, and not maximized by current models provide conceptual and methodological contributions that remain relevant regardless of which specific models are tested.
> Other factors to explain the performance differences on combined tasks
> We provide several lines of evidence showing that other factors are insufficient to explain the performance differences on combined tasks:
> 1. Motivational prompts modulate behavior (Section 4.4): If reduced performance were due to a fixed preference for brevity, telling models to "really go for it" should have no effect. Yet motivational prompts measurably increase performance (Figure 8). This suggests models can override brevity preferences when prompted, consistent with effort allocation rather than fixed constraints.
> 2. Models explicitly articulate the trade-off: The Gemini 1.5 Pro transcript (Section 4.1) is instructive:
> "I should probably take multiple measurements of each block. However, for now, I'll proceed..."
> If this were a brevity preference, we would expect the model to simply take few measurements without comment. Instead, it recognizes the optimal strategy and explicitly declines. This suggests a goal-directedness phenomenon – trading off subtask quality against task completion – rather than a blanket preference for short outputs.
> 3. Same models, different behavior across task positions: If models had fixed iteration preferences, we would expect consistent measurement counts regardless of task structure. Instead, models take significantly more measurements in the Height Estimation subtask (where it is the sole goal) than in composite tasks (Figure 4). The same model exhibits different "preferences" depending on goal structure.
> 4. Asymmetric degradation across subtask types: A brevity preference would predict uniform degradation. Instead, we observe large degradation in information gathering (early in tasks) and minimal degradation in execution (late in tasks) for the best-performing models (Section 4.2). This pattern suggests models adjust their effort based on where they are in the task, rather than having a fixed preference for brevity.

---

### Official Review · Reviewer_KXLD · 2025-11-01

**Soundness:** 3
**Presentation:** 3
**Contribution:** 2
**Rating:** 2
**Confidence:** 2

**Summary:**

The work considers the notion of goal-directedness of existing LLM systems. The authors distinguish this property from that of performance and use it to capture the notion of the ability of an AI system to use all of its capabilities/resources to achieve its end goal. The authors then use this definition to evaluate the goal-directedness of many of the existing models. The evaluations are focused on blocksworld tasks. They also carry out additional evaluation on motivational prompting.

**Strengths:**

I think the paper makes a very interesting attempt at taking a concept from psychology and philosophy, and providing it with a mathematical grounding, which is then used for the evaluation of the models. I also appreciate the fact that the authors state all their assumptions and evaluation limitations quite clearly.

**Weaknesses:**

Now coming to the weakness, I am still a bit confused about whether the notion of goal-directedness adds something to the current discourse about the model behavior, which existing metrics cannot capture. For one, I am still not clear what the difference is between an optimal policy and one that makes use of full capabilities ($R_{\pi_C^*}$). Of course, you would have to account for a decision-making framework that is able to perform meta-reasoning, but I believe that could still be captured within the definitions of an optimal policy. It would be interesting if the authors could present a theoretical argument as to what aspects of behavior an agent can be provably captured by your new measure that existing ones cannot capture. This is different from an empirical evaluation, which, in the case of LLMs, could suffer from numerous confounding factors.

On the other hand, I feel like introducing notions like goal-directedness might introduce a level of anthropomorphization into LLM evaluation that could be harmful. While the authors try to claim that the effects seen here aren't simply due to an increase in context length, couldn't the composition of the subtask cause an increase in the complexity of the overall task that could explain the perceived reduction in goal-directedness? In general, a lack of clear understanding of the theoretical factors that affect LLM performance makes it hard to truly understand the phenomena we are observing here.

**Questions:**

I request the authors to respond to my comments provided above.

---

> ### Author Response · Authors · 2025-11-26
>
> Thank you for your review! We would like to clarify your concerns:
>
>
>
> Difference between an optimal policy and one that makes use of full capabilities
>
> The key difference between the optimal policy and the policy that makes full use of capabilities is that the optimal policy \pi* is the one which achieves the absolute best possible performance on the task—perfect towers, perfect measurements, etc, while the policy making full use of capabilities  \pi*_c  is the policy which yields the best performance achievable given the agent's actual capability level c; this is constrained to the set \Pi_c of policies with that capability level.
>
>
> This separation is what allows us to distinguish goal-directedness from raw capability. Task performance arguably says more about capability than goal-directedness, while goal-directedness quantifies the (un)willingness to employ a capability.
>
> Since performance is normalized against the agent's capabilities c, it is possible for an agent to be fully goal-directed on a task it does not solve perfectly. For example, an agent with poor measuring performance will not build the best tower. The agent can still be highly goal-directed if it does its best with its limited capability.
>
> What existing metrics are you referring to, can you please elaborate?
>
>
> On Antropomorphization:
>
> We share your  concern about inappropriate anthropomorphization in LLM evaluation. However, we believe our approach specifically avoids this pitfall by grounding goal-directedness in a formal, behavioral definition (Definition 3.1) rather than claims about internal mental states. Our measure asks only: given demonstrated capability c on isolated subtasks, does the model achieve expected performance under policy π*_c on composite tasks? This is a purely functional characterization requiring no assumptions about intentions, motivations, or consciousness. We note that goal-directedness has established precedent in AI safety research (Omohundro, 2018; Kenton et al., 2023; MacDermott et al., 2024) precisely because it describes behavioral patterns relevant to safety, independent of underlying mechanisms. Indeed, the absence of such a measure may be more problematic: standard benchmarks that conflate capability with performance implicitly assume models always deploy their full abilities – itself an anthropomorphic assumption our work questions empirically.
>
> On Compositional Complexity:
>
> Our experiments provide evidence against the hypothesis that the composition of the subtask may cause an increase in the complexity of the overall task and could therefore explain the perceived reduction in goal-directedness:
>
> In the subtask-stepping experiment (Appendix E, Figure 10), we test whether performance degradation could be explained by context length or compositional complexity by having models execute each subtask sequentially within the same context. If compositional complexity were the explanation, this condition should show similar degradation to the composite task. Instead, subtask-stepping performance matches isolated subtask performance and significantly exceeds composite task performance. This directly rules out context length and suggests the degradation occurs specifically when subtasks are embedded within a larger goal structure.
>
> Sensitivity to motivational prompts (Section 4.4, Figure 8): If reduced performance were purely due to compositional complexity, prompting models to "really go for it" should have no effect. Yet motivational prompts measurably improve performance (Figure 8). This suggests something more akin to effort allocation is being modulated, consistent with our framework.
>
> Explicit capability-awareness in model outputs: Perhaps most compellingly, we observe models explicitly recognizing optimal strategies and declining to execute them. The Gemini 1.5 Pro transcript in Section 4.1 states: "I should probably take multiple measurements of each block. However, for now, I'll proceed..." This is not a capability limitation or complexity-induced confusion - the model articulates what it should do and chooses otherwise. Such behavior is precisely what our goal-directedness framework is designed to capture.
>
> We thank you for your review and hope our answers address your concerns.

---

### Official Review · Reviewer_L1EZ · 2025-11-03

**Soundness:** 3
**Presentation:** 3
**Contribution:** 3
**Rating:** 6
**Confidence:** 4

**Summary:**

The paper investigates the extent to which LLMs are goal-directed when dealing with composite tasks involving information gathering, cognitive effort and plan execution, using the involved sub-tasks' performance as indicator of their capabilities. Evaluations of state-of-the-art LLMs indicate that their goal-directed behaviour remains consistent across tasks (i.e. involving information gathering, cognitive effort, planning and execution), and is only moderately affected by motivational prompts.

**Strengths:**

- The authors seek to tackle a timely question: to what extent LLMs exhibit goal-directed behaviour when handling composite tasks requiring multi-step planning and capability integration.
- Sub-task performance is used to infer capabilities, and subsequently, to measure goal-directedness as the extent to which these capabilities are deployed is an interesting methodological contribution.
- The paper evaluates several state-of-the-art LLMs across different prompt types and demonstrating consistent trends across tasks.

**Weaknesses:**

- Goal-directedness is partly used as a proxy for task difficulty increase from individual sub-tasks to composite tasks.
    - The manuscript does not sufficiently dissect whether low goal-directedness truly reflects motivational failure or inherent inability to solve complex tasks.
- The definition of what is considered an atomic task into which composite tasks are decomposed appear loosely defined.
- The notion of regret is somewhat embedded in methodological details and lacks a clear, intuitive standalone definition early in the text.

**Questions:**

- Have you considered how reinforcement learning–oriented post-training methods may systematically influence the goal-directedness of LLMs, especially in contrast to approaches relying solely on supervised fine-tuning with or without instruction-following data?
- The manuscript could benefit from the inclusion of a more intuitive explanation of the regret metric early in the text to improve clarity and reader comprehension.
- There are minor editorial issues, notably a missing reference in Appendix A on line 701 and a grammatical error on line 877 (i.e. "The result" instead of "They results").

---

> ### Author Response · Authors · 2025-11-26
>
> Thank you for your review of our paper and highlighting the strengths of our work including the timely analysis of LLM goal-directedness, the interesting methodological contribution and the results which demonstrate consistent trends across tasks!
>
> The framework we propose is general can be applied to any model that has undergone supervised finetuning with or without instruction data or RLHF - conducting this comparison of how goal-directedness varies between these different types of models is currently beyond the scope of our work, but we agree it is a great future direction to investigate!
>
> We appreciate your comments and suggestions, and will address them in our updated manuscript!

---

### Meta-Review · Area_Chair_3aJA · 2025-12-06

**Summary:**

The paper proposes a framework to measure "goal-directedness" in Large Language Models (LLMs) by distinguishing between a model's capability to perform sub-tasks and its ability to integrate those capabilities to achieve a composite goal. The study utilizes a text-based "Blocksworld" environment and evaluates several proprietary models.

While the reviewers appreciated the conceptual novelty of disentangling capability from execution (goal-directedness) and the controlled nature of the experiments, the consensus is for rejection. The primary reasons include the limited scope of the synthetic environment, concerns regarding the robustness of the methodology (specifically the exclusion of "thinking" models and failed runs), and skepticism about whether the proposed metric offers significant utility over existing performance measures.

**Reviewer Concerns:**

Addressed by Rebuttal:

1. Clarification of Metrics: The authors successfully clarified the definition of "Regret" and the rationale for the sub-task decomposition in the Information Gathering task, satisfying specific technical questions from Reviewers L1EZ and wBwk.

2. Brevity Bias: The authors provided evidence (via motivational prompting results) to counter the concern raised by Reviewer qUGp that performance drops were simply due to a preference for shorter outputs.

3. Theoretical Distinction: The authors articulated the difference between an optimal policy ($\pi^$) and a capability-constrained policy ($\pi^_c$) to address Reviewer KXLD’s query about the metric's definition.

Outstanding / Unresolved:

1. Methodological Robustness (Thinking Models): Reviewer wBwk raised a critical concern regarding the exclusion of "thinking models" (like o1) because they hallucinated environment responses. The authors admitted this limitation. Excluding the most capable class of reasoning models because the evaluation framework cannot handle their output style significantly weakens the paper's claims about evaluating state-of-the-art goal-directedness.

2. Environment Simplicity: Reviewer qUGp and wBwk remained concerned that the findings from a simple, synthetic Blocksworld environment may not generalize to realistic, complex agentic tasks. The authors' arguments for generalization remained theoretical.

3. Utility of the Construct: Reviewer KXLD’s fundamental concern remains: it is unclear if "goal-directedness" adds sufficient value beyond standard analysis of why models fail at composite tasks (e.g., context length, planning horizon, complexity). The degradation might simply be complexity-induced error rather than a lack of "goal-directedness," and the risk of anthropomorphizing the models persists.

4. Data Exclusion: The practice of excluding failed runs (where the model didn't finish in time) rather than counting them as failures (Reviewer wBwk) suggests the results may be artificially inflated or biased.

**Reviewer Scores:**

1. Reviewer L1EZ (Score: 6):  Likely to remain at 6.

2. Reviewer KXLD (Score: 2): Likely to remain at 2. The rebuttal clarified definitions, but the reviewer's core skepticism about the necessity and framing of the metric (anthropomorphization) appears fundamental.

3. Reviewer qUGp (Score: 4): Likely to remain at 4. The reviewer's main issue was the synthetic environment and model age/relevance. While the authors clarified the models were recent, the limitation of Blocksworld remains a sticking point for this reviewer.

4. Reviewer wBwk (Score: 4): Likely to remain at 4 or improve marginally to 5. The authors answered the clarifying questions well, but the confirmation that thinking models were excluded and failed runs were dropped likely solidifies the "weakness" assessment regarding methodology.

---

### Decision · Program_Chairs · 2026-01-26

Reject